

# Spatial heterogeneity in the exclusive use of hygienic materials during menstruation among women in urban India

Aditya Singh[*] and Mahashweta Chakrabarty[*]

Department of Geography, Banaras Hindu University, Varanasi, Uttar Pradesh, India
[*] These authors contributed equally to this work.

## ABSTRACT

**Background**. Menstrual hygiene is essential for women to live with dignity. However, a large proportion of Indian women still suffer from unhygienic menstrual practices leading to reproductive tract infections. To understand the socioeconomic and bio-demographic determinants of menstrual hygiene practices, various national or local level studies have been conducted in India and around the world, however, no previous study has tried to understand the spatial heterogeneity across Indian districts in the use of hygienic materials among young urban women.

**Methods**. This study used data from 54,561 urban women aged 15–24 from the National Family Health Survey-5. Global Moran's I was applied to assess the degree of spatial autocorrelation and cluster and outlier analyses to locate hot-spots and clod-spots in the exclusive use of hygienic materials across the districts. Ordinary least square, spatial lag, and error models were used to identify determinants of exclusive use of hygienic materials.

**Results**. Approximately 66.8% of urban women exclusively use of hygienic materials which varied across districts. Global Moran's I of 0.46 indicated positive spatial autocorrelation in the outcome. Cluster and outlier analysis revealed cold-spots in central Indian districts and hotspots in south Indian districts. Results of spatial error model identified women's years of schooling, marital status, social group, and household wealth were major determinants of the exclusive use of hygienic materials among urban women across Indian districts.

**Conclusion**. Substantial spatial heterogeneity in the outcome among urban women in India suggests the need to design targeted and context-specific behavioural interventions and programs for women in urban India.

Corresponding authors
Aditya Singh, adityasingh@bhu.ac.in
Mahashweta Chakrabarty, mahashweta.c1997@gmail.com

## INTRODUCTION

Developing countries including India are witnessing a rapid growth in their urban population. Between 1951 and 2020, India's urban population has doubled, from 17.0% to 34.9% (285 million), with an annual increase of 2.3 percent (*United Nations, 2018*).
For the first time since independence, India's urban population grew at a higher rate than its rural population, with a 31.8 percent increase between 2001 and 2011 (*Chakravarthy, Rajagopal & Joshi, 2019*). Although cities are the engines of economic growth and they bring significant improvements in health and living standards of the urban residents, nevertheless the cities demonstrate higher level of inequality in health outcomes than rural areas (*McMichael, 2000*). Evidence suggests that with rapid urbanisation in India, essential public services demand more attention to address emerging challenges (*Chakravarthy, Rajagopal & Joshi, 2019*).

Inadequate access to toilets, sanitation, water, and drainage, as well as growing poverty and a lack of disposable income in cities, makes living conditions in Indian cities unsanitary and unhygienic (*Mberu et al., 2016*; *Chakravarthy, Rajagopal & Joshi, 2019*; *Vogel, Hwang & Hwang, 2022*). During the COVID-19 pandemic, the demand for water, sanitation, and hygiene (WASH) had increased to a large extent, particularly in cities (*Yamakoshi, 2020*; *Ingraham, Sharma & Joe, 2021*). Consequently, the poor and less educated women's menstrual health and hygiene needs went unmet, due to the limited WASH resources available during the pandemic (*Yamakoshi, 2020*). Typically, in Indian households, it is women who are expected to balance between household's income and needs (*Sharma et al., 2020*; *Ingraham, Sharma & Joe, 2021*). In the process, the needs and demands of other household members are prioritised and women have to compromise their own health and hygiene needs (including menstrual) (*Chakravarthy, Rajagopal & Joshi, 2019*). Neglecting menstrual health and hygiene can have an adverse effect on a woman's reproductive and gynaecological health (*The World Bank, 2022*). Reproductive and gynaecological diseases are the fifth leading cause of reduced disability-adjusted life years (DALY) in the age group 10–24 years, after mental health-related diseases (*Vos et al., 2020*). Evidence suggests that proper menstrual hygiene management can reduce the burden of reproductive and gynaecological diseases to a considerable extent (*Roeckel, Cabrera-Clerget & Yamakoshi, 2019*). Therefore, there is a need to address menstrual health and hygiene needs of women, especially in the low- and middle-income countries where access to safe and effective means of managing menstrual hygiene is limited. Since the challenges surrounding menstrual health and hygiene have massive implications for women's/girls' health, education, gender, and WASH, it is argued that it would be difficult to achieve the sustainable development goals (SDGs) without addressing these challenges (*Sommer et al., 2021*).

To prevent menstrual bloodstains from being visible women use various materials like clothes, sanitary napkins, tampons, menstrual cups, or others. In general, these material grouped into two categories: hygienic materials (sanitary pads, tampons, menstrual cups) and unhygienic materials (clothes, old rags, socks, newspaper, dried leaves, etc.) (*Anand, Unisa & Singh, 2015*). Research in low- and middle-income countries reveals a widespread trend of urban women and girls relying on unhygienic materials such as clothing during their menstrual periods due to a lack of affordable menstrual care products and proper disposal facilities (*Van Eijk et al., 2016*; *Elledge et al., 2018*). This inadequate management of menstruation poses a significant risk to their health, leading to a heightened probability of infections and complications related to the reproductive and urinary tracts, including gynaecological diseases. (*Das et al., 2015*; *Torondel et al., 2018*; *Almeida-Velasco*

& Sivakami, 2019). The inability to manage menstruation in a hygienic and dignified manner also infringes upon their basic human rights, including access to education, health, and a healthy environment, as well as water and sanitation at work. As a result, menstrual health constitutes both a crucial human rights issue and a serious public health concern that requires immediate attention to achieve the Sustainable Development Goals in an equitable and consistent manner (Hennegan, 2017; Babbar et al., 2022).

Much of the current literature on menstrual hygiene practices among Indian women have reported that there are differences in women's choice of menstrual materials based on geography, biodemographic characteristics, socioeconomic status, and mass-media exposure (Misra et al., 2013; Paria, Bhattacharyya & Das, 2014; Kathuria & Raj, 2018; Goli et al., 2020; Ram et al., 2020; Babbar, Saluja & Sivakami, 2021; Chauhan et al., 2021; Roy et al., 2021). A number of studies have examined the link between poor menstrual hygiene management and its adverse health outcomes (Anand, Singh & Unisa, 2015; Almeida-Velasco & Sivakami, 2019). Several studies in the recent past made an effort to investigate the knowledge, attitudes, and practices regarding hygienic material use among adolescent girls in India (Misra et al., 2013; Balamurugan, Shilpa & Shaji, 2014; Kansal, Singh & Kumar, 2016; Van Eijk et al., 2016; MacRae et al., 2019; Das & Majhi, 2020; Ramaiya & Sood, 2020). However, studies addressed hygienic menstrual practices among urban women in India are few and far between. Most of the existing studies are small-scale studies, i.e., they have covered a limited geographic area. None of them have attempted to identify the spatial variability in the usage of hygiene materials during menstruation among urban women across the district of India (Van Eijk et al., 2016; Rajagopal & Mathur, 2017; Wagh, Upadhye & Upadhye, 2018; McCammon et al., 2020). The dearth of research on the menstrual health and hygiene of urban women in India is due to the widespread notion of 'urban advantage' which states that urban women enjoy better health than their rural counterparts. While the exclusive use of hygienic materials during menstruation in urban India has risen considerably over last few years, a considerable number of women still use unhygienic materials.

In order to achieve universal use of hygienic materials during menstruation among women in urban India, it is essential for policymakers to comprehend the geographical disparities in the exclusive use of such materials. However, there is limited research that analyzes the spatial heterogeneity in the exclusive use of hygienic material among women at the district level in urban India and the factors that contribute to such geographical variations. This study attempts to fill this gap in order to furnish policymakers with valuable insights to advance the widespread use of hygienic materials during menstruation among women in urban India.

This study differs from the previous studies in three key areas. Firstly, it aims to identify the Indian districts where the exclusive use of hygienic materials among the urban women is significantly clustered and adequately deficient. Secondly, it investigates the factors behind such spatial disparity in India in the exclusive use of hygienic materials among urban women aged 15–24. Thirdly, it is based on the latest nationally representative data from the National Family Health Survey-5, which shall help policymakers with latest evidence on of exclusive use of hygienic materials and regional inequality in the same. Thus, this

work presents novel analyses of district-level geospatial patterns, spatial associations, and correlates of exclusive use of hygienic materials during menstruation among urban women in India.

# DATA AND METHODS

## Ethics statement

Ethical permission was not required as the data is taken from an open domain. The National Family Health Survey-5 dataset used in the study is publicly available at the official website of Demographic and Health Surveys (DHS) (https://dhsprogram.com/data/available-datasets.cfm). Anyone can obtain this data by making a formal request to the DHS.

## Data source

We used data from the most recent fifth round of the National Family Health Survey (NFHS-5), 2019-21. The NFHS provides data on a variety of demographic, socioeconomic, maternal and child health outcomes, morbidity and healthcare, reproductive health, and family planning concerns (*International Institute for Population Sciences, 2020*). The NFHS-5 national report contains detailed information on the sampling procedures used in the survey (*International Institute for Population Sciences, 2020*). A total of 724,115 women aged 15–49, from 636,699 households were interviewed for the NFHS-5. The research included 54,561 urban women aged 15–24 from 28 states, eight union territories, and 707 districts of India. (See Fig. 1 for sample selection process).

## Dependent variable

In the NFHS-5, a multiple-choice question was asked to women on the material that they used during their menstruation. There were seven options for response: cloth, sanitary napkins, locally made napkins, tampons, menstrual cups, nothing, and others. On the basis of these responses, we created a binary outcome variable titled 'exclusive use of hygienic materials', in which women using materials such as sanitary napkins, locally made napkins, tampons, and menstrual cups were coded as ''1''; and those using non-hygienic and reusable materials such as cloths, both hygienic and non-hygienic materials, or not using any form of menstrual materials were coded as ''0'' (*Anand, Singh & Unisa, 2015*; *Ram et al., 2020*; *Vishwakarma, Puri & Sharma, 2020*; *Roy et al., 2021*; *Singh et al., 2022a*; *Singh et al., 2022c*; *Singh et al., 2022b*).

## Independent variables

The choice of independent variables was guided by existing literature on menstrual hygiene management (*Anand, Singh & Unisa, 2015*; *Kathuria & Raj, 2018*; *Ram et al., 2020*; *Vishwakarma, Puri & Sharma, 2020*; *Chauhan et al., 2021*; *Roy et al., 2021*). To illustrate how different factors are associated with women's exclusive usage of hygienic materials, a conceptual framework has been adopted from a previous study (*Singh et al., 2022b*). We considered a range of demographic predictors such as current age of respondents (in years), age at menarche (in years) and socioeconomic predictors such as age at marriage (married before 18 years, and married after 18 years or not-married), respondent's years
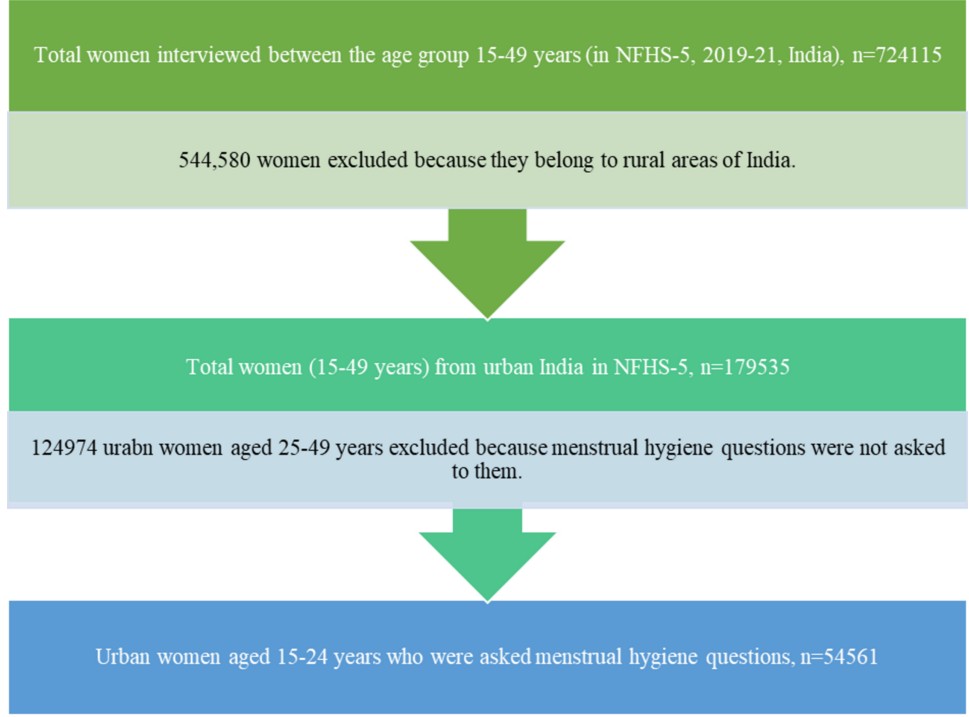

**Figure 1** Flow chart showing the process of selection of urban women (15–24 years) sample for the current study.

of schooling (in years), religion (Hindu, non-Hindu), social groups (SC/ST, non-SC/ST), household wealth status (poor, non-poor), and mass-media exposure (no mass-media exposure, exposure to any of the mass-media). Detailed information and categorization of variables is given in Table 1.

## Statistical analysis

To examine the regional variation in the exclusive use of hygienic materials, we first prepared a district-level map depicting the proportion of urban women reporting exclusive use of hygienic materials. Further, to determine the degree of spatial autocorrelation in our spatial data we calculated Global Moran's I. Moran's I values range between −1 and +1 (*Singh et al., 2022a*). While −1 indicates perfect clustering of dissimilar values (perfect dispersion), +1 indicates perfect clustering of similar values (it is the opposite of dispersion). A Moran's I value of 0 indicates perfect randomness (*Anselin, 1995*).

Later to detect the significant clusters and outliers of high and low prevalence of exclusive use of hygienic materials among young urban women in India, we conducted cluster and outlier analysis (Anselin Local Moran's I) (*Roquette, Nunes & Painho, 2018*). This analysis identifies two types of clusters: (a) high–high clusters (districts with high values that are surrounded by districts with similar high values, often known as hotspots) and (b) low–low clusters (districts with low values surrounded by other districts with similarly low values, often referred to as cold-spots) (*Singh et al., 2022a*). This method also generates two types

**Table 1  Operational definition of variables used in the study.**

| Variables | Description |
|---|---|
| Age of respondent (in years) | Age of respondent indicates the current age of women at the time of interview. For spatial analysis, we have taken the mean age of each district. |
| Age at menarche (in years) | Age at menarche is the age when a girl gets her first menstruation. For spatial analysis, we have considered mean age at menarche of each spatial unit. |
| Respondent's years of schooling | Respondent's years of schooling has been obtained from the years of schooling variable of NFHS-5. |
| Marital status | By subtracting the year of birth from year of marriage we have first calculated the age of marriage of any respondent. Those women who were unmarried were coded as 0 and labelled 'not married'. Women who were married before 18 years of age is coded as 1 and labelled 'married before 18 years'; and women who were married after 18 years are coded as 2 and labelled 'married after 18 years'. For spatial regression, we have taken the proportion of women who were married before 18 years. |
| Religion | In NFHS-5, there are different types of religion *i.e.*, Hindu, Muslim, Christian, Sikh, Buddhist, Jain, Jewish, Parsi /Zoroastrian etc. For the purpose of this study, religion has been recoded into two categories Hindu (1) and non-Hindu (0). |
| Social groups | The Government of India has identified a total of four social groups *i.e.*, Scheduled Caste (SC), Scheduled Tribe (ST), Others Backward Classes (OBC) and others (General). For the suitability of our study, we merged SC and ST into the first sub-category which is SC & ST (1) and we merged the rest two social classes into a second sub-category *i.e.*, non-SC/ST (0). |
| Household wealth | Household wealth is a composite index of household amenities and assets. NFHS-5 classified it into five categories *i.e.*, poorest; poorer; middle; richer; richest. For suitability of our study, we recode this variable into two categories. The first is poor (1) which comprises the first two quintiles *i.e.*, poorest and poorer; second is non-poor (0) which comprises the last three quintiles *i.e.*, middle, richer and richest. |
| Mass media exposure | Three questions were asked to women in NFHS-5 survey. They are (i) how often they read newspaper/magazines, (ii) how often they watch television, and (iii) how often they listen to radio. The responses are 'almost every day', at least once a week, less than once a week and not at all. Based on these responses, we have formed a dichotomous variable. Women were considered 'have' (1) mass media exposure if they had exposure to any of these sources and as 'no exposure' (0) if they responded with 'not at all' for all three sources of media |

of outliers: (a) high–low outliers (districts with high values that are surrounded by districts with low values) and (b) low–high outliers (districts with low values that are surrounded by high-value neighbours) (*Singh et al., 2022a*).The study employs Queen Contiguity weights

(contiguity edges and corners), which represent whether or not geographical units share edges and corners (*Anselin & Rey, 2014*).

Finally, we fit a series of regression models to investigate the relevant correlates of the exclusive use of hygienic materials among urban women at district level in India. The unit of analysis was district. We proceeded by conducting an OLS regression wherein we also implemented Lagrange Multiplier (LM) and Robust Lagrange Multiplier (RLM) diagnostic tests to detect spatial dependence in our dataset (*Thomas & Aryal, 2021*). If any of the LM or RLM tests become significant, one should run the spatial lag model (SLM) or spatial error model (SEM) and check which model is best fit for their dataset. The explanation on why OLS, SLM and SEM was conducted, and how to check its significance is given below in the flow chart (see Fig. 2) (*Anselin & Rey, 2014*; *Thomas & Aryal, 2021*).

In our study, both the LM and RLM tests for SLM and SEM were found to be significant. Since the SEM had the lowest Akaike Information Criterion (AIC) value, highest log-likelihood value, highest LM, and Robust LM values, we estimated SEM to be the best model out of the three. ArcGIS 10.5, GeoDa, GeoDaSpace, and Stata17 were utilised to analyse the data for this study (*StataCorp, 2019*; *Anselin, Syabri & Kho, 2009*; *ESRI, 2011*).

# RESULTS

## Sample characteristics

Table 2 shows the background characteristics of 54,561 sampled women aged 15–24 years from urban India. About 53.4% of the sampled women had schooling for 11 years and above. Only 3.4% of women never went to school. Approximately 30.0% of the women in the sample were SC/ST, and about 70.0% were Hindu. Around a quarter of women were from poorest household, and majority of sampled women had at least one mass media exposure.

## Exclusive use of hygienic materials by background characteristics

Table 3 presents the proportion of urban women exclusively using hygienic materials by their background characteristics. The prevalence of exclusive use of hygienic materials among urban women who had schooling of 11 years or above (75.9%) was almost double than those who never attended school (33.4%). While 86.0% of women in richest households used hygienic materials, only 48.0% of women in poor households used such materials. Prevalence of using hygienic materials was slightly higher in Hindu (70.3%) women as compared to other religions (61.5%). Exclusive hygienic material use was significantly higher among women exposed to mass media (70.1%) as compared to those who reported no exposure to mass media (44.8%).

## Spatial patterns of exclusive use of hygienic materials among urban women across 707 districts of India

The state-level analysis gives a generalised pattern of spatial variation in the exclusive use of hygienic materials, but it fails to reveal any meaningful spatial heterogeneity. Therefore, to unravel the granular spatial patterns, we created a prevalence map of exclusive use of hygienic materials in across 707 districts of India.
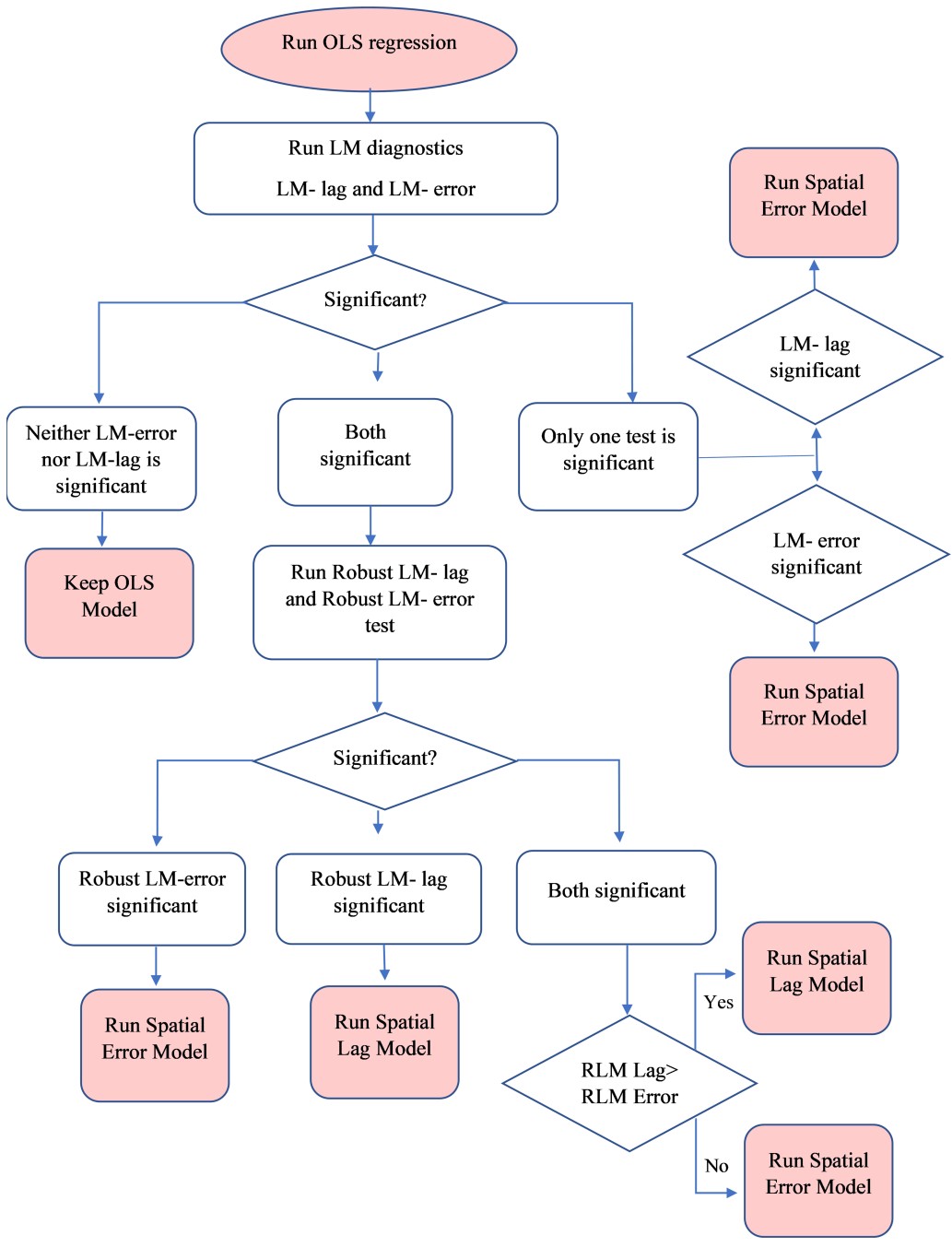

**Figure 2** Flowchart of selecting spatial lag model or spatial error model according to Lagrange multiplier or Robust Lagrange multiplier tests.

Figure 3 shows that out of 707 districts, exclusive use of hygienic materials were less than 50% in 159 districts; between 50–80% in 356 districts, and above 80% in the remaining 183 districts. The lowest exclusive use of hygienic material (below 30%) was identified in the districts of Lakhisarai, Nawada, Purba Champaran, Sitamarhi, Khagaria, Gopalganj,

**Table 2  Percentage distribution of urban women aged 15–24 years by background characteristics, NFHS-5 (2019-21), India.**

| Background characteristics | N (54,561) | Weighted % |
|---|---|---|
| **Respondent's current age (in years)** | | |
| 15–19 | 26,509 | 48.59 |
| 20–24 | 28,052 | 51.41 |
| **Age at menarche (in years)** | | |
| Less than 13 | 10,871 | 20.06 |
| 13 and above | 43,321 | 79.94 |
| **Respondent's years of schooling** | | |
| No schooling | 1,847 | 3.39 |
| Less than 5 years | 2,185 | 4.00 |
| 6–10 years | 21,415 | 39.25 |
| Above 11 years | 29,114 | 53.36 |
| **Marital status** | | |
| Not married | 40,375 | 74.00 |
| Married before 18 years old | 3,807 | 6.98 |
| Married after 18 years old | 10,379 | 19.02 |
| **Religion** | | |
| Hindu | 38,368 | 70.32 |
| Muslim | 10,564 | 19.36 |
| Christian | 3,573 | 6.55 |
| Others | 2,056 | 3.77 |
| **Social groups** | | |
| Non-SC/ST | 38,275 | 70.15 |
| SC/ST | 16,286 | 29.85 |
| **Household wealth** | | |
| Poorest | 12,857 | 23.56 |
| Poorer | 12,277 | 22.50 |
| Middle | 10,985 | 20.13 |
| Richer | 10,006 | 18.34 |
| Richest | 8,436 | 15.46 |
| **Exposure to mass media** | | |
| No mass media exposure | 4,499 | 8.25 |
| At least one mass media exposure | 50,062 | 91.75 |

**Notes.**

N, Sample size

Kaimur of Bihar, Bara Banki, Lalitpur, Sant Kabir Nagar, Mau, Shrawasti, Ballia, Hardoi, Ambedkar Nagar of Uttar Pradesh, Chirang, Hojai, Nagaon, Lakhimpur, Karimganj of Assam, Balaghat, Khandwa of Madhya Pradesh, Balod, Mahasamund, Raipur, Bemetara of Chhattisgarh, Baramula, Badgam, Kupwara, Anantnag of Jammu and Kashmir, Pakur of Jharkhand, Hamirpur of Himachal Pradesh. In contrast, districts in Tamil Nadu, Telangana, Andhra Pradesh, Punjab, Haryana, Sikkim, Arunachal Pradesh, and Mizoram have higher exclusive use of hygienic materials. Two lagging districts in terms of exclusive

**Table 3** Percentage of urban women aged 15–24 years who exclusively used hygienic materials for menstrual bloodstain prevention, by selected background characteristics, NFHS-5, 2019-21.

| Background characteristics | Exclusive use of hygienic materials (weighted %) N=54561 | 95% CI | |
|---|---|---|---|
| | | Lower | Upper |
| **Respondent's current age** (in years) | | | |
| 15–19 | 68.93 | 67.75 | 70.09 |
| 20–24 | 67.30 | 66.21 | 68.37 |
| **Age at menarche** (in years) | | | |
| Less than 13 | 69.17 | 67.57 | 70.72 |
| 13 and above | 68.06 | 66.99 | 69.11 |
| **Respondent's years of schooling** | | | |
| No schooling | 33.35 | 29.73 | 37.17 |
| Less than 5 years | 39.61 | 36.32 | 43.01 |
| 6–10 years | 63.06 | 61.75 | 64.35 |
| Above 11 years | 75.87 | 74.91 | 76.80 |
| **Marital status** | | | |
| Not married | 70.98 | 69.94 | 72.00 |
| Married before 18 years old | 52.36 | 49.80 | 54.90 |
| Married after 18 years old | 63.93 | 62.41 | 65.43 |
| **Religion** | | | |
| Hindu | 70.32 | 69.29 | 71.32 |
| Muslim | 56.31 | 54.05 | 58.54 |
| Christian | 80.37 | 77.08 | 83.29 |
| Others | 83.79 | 80.86 | 86.34 |
| **Social groups** | | | |
| Non-SC/ST | 69.08 | 67.99 | 70.16 |
| SC/ST | 65.14 | 63.54 | 66.72 |
| **Household wealth** | | | |
| Poorest | 48.04 | 46.11 | 49.98 |
| Poorer | 64.21 | 62.58 | 65.80 |
| Middle | 70.84 | 69.32 | 72.32 |
| Richer | 77.44 | 76.00 | 78.82 |
| Richest | 86.02 | 84.76 | 87.18 |
| **Exposure to mass media** | | | |
| No mass media exposure | 44.79 | 42.06 | 47.55 |
| At least one mass media exposure | 70.07 | 69.10 | 71.01 |

**Notes.**

N, sample size; CI, confidence interval.

use of hygienic material materials (Kaimur and Ambedkar Nagar) were from Bihar and Uttar Pradesh where the prevalence was less than 15% respectively.

## Global Moran's I result

To detect the presence of spatial autocorrelation, we have computed Global Moran's I value for the exclusive use of hygienic materials at the district level in India. The Global Moran's I index is 0.46 with the Z score of 19.88 and a *p*-value of 0.00, so there is less than a

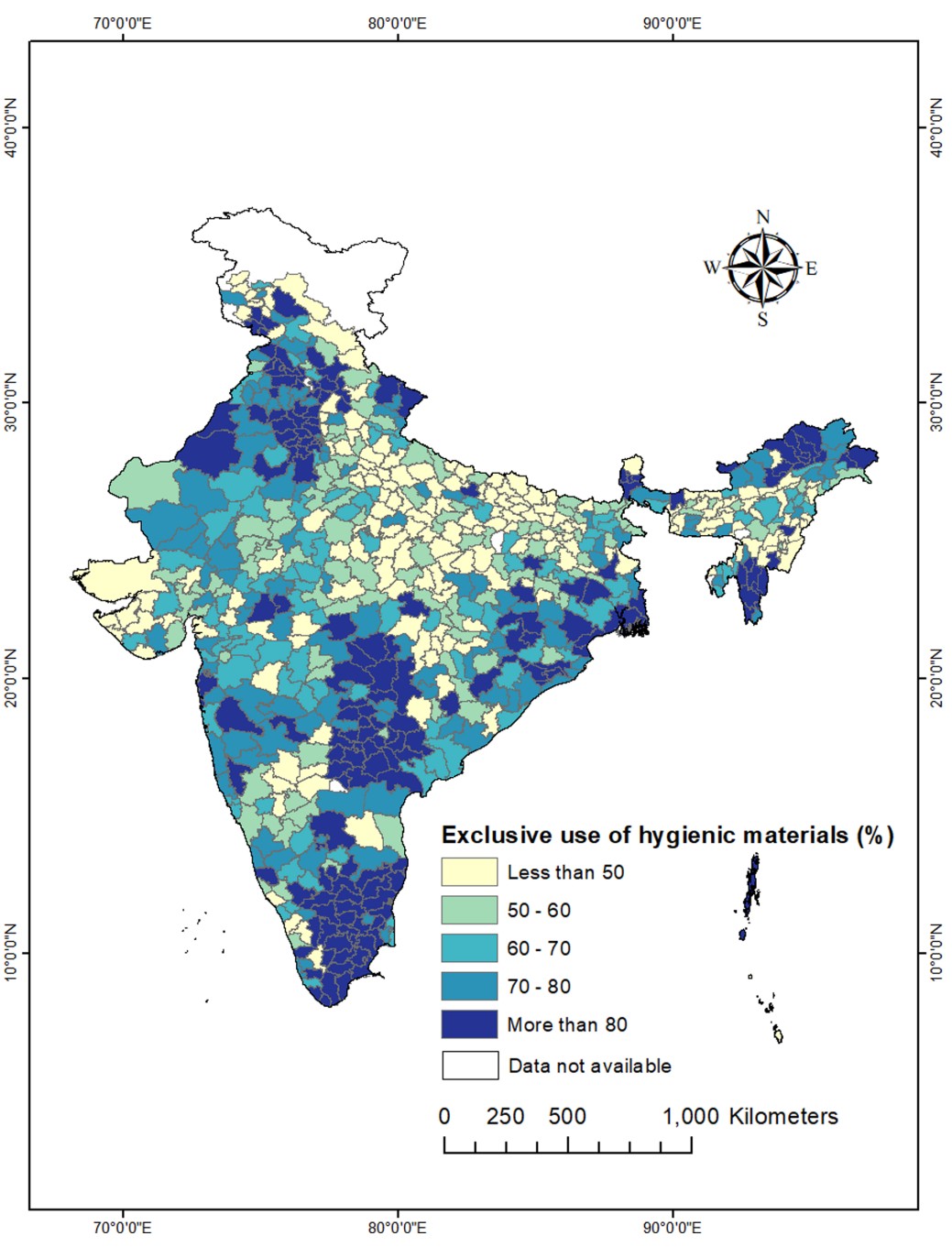

**Figure 3** District-wise distribution of exclusive use of hygienic materials during menstruation among urban women (15–24 years) in India, NFHS-5, 2019-21.

1% likelihood that this clustering results from random chance, which signifies high spatial autocorrelation. Consequently, it becomes essential to understand the spatial clustering of exclusive use of hygienic materials at the local (district) level. As Global Moran's I cannot locate where the clustering is; we have computed local indicators of spatial association, *i.e.,* cluster and outlier analysis.

## Cluster and outlier analysis (Anselin Local Moran's I)

Figure 4 shows statistically significant spatial outliers, hot-spots, and cold-spots. High-high clusters or hot-spots were identified in the southern states of India, particularly in most of the districts of Tamil Nadu, Telangana, and Andhra Pradesh. Hot-spots were discovered in the districts of Punjab, Haryana, Himachal Pradesh, northern Rajasthan. On the other hand, majority of the districts of Uttar Pradesh, Bihar, Madhya Pradesh, and northern Chhattisgarh showed statistically significant low-low clustering. North-eastern states found to have some cold-spots, mainly districts of Meghalaya, Assam, Manipur, and Nagaland.

## Ordinary least squares (OLS) and spatial regression models

Global and Local Moran's I index indicated strong statistically significant geospatial clustering in the outcome variable, so we used ordinary least squares (OLS), the spatial lag model and the spatial error model to detect relevant determinants of exclusive use of hygienic materials.

The OLS, SLM, and SEM results are presented in Table 4. Without taking into account the data's spatial structure, OLS estimation was used to check the association between the exclusive use of hygienic materials and their correlates. Two sets of LM and robust LM tests were used during OLS estimation to determine the model's suitability for predicting spatial dependence in our spatial dataset. We decided to run both SLM and SEM in our analysis because both LM and robust LM values for SLM and SEM were statistically significant, indicating spatial dependence in our dataset. During the comparison, we highlighted that the robust LM value of lag was greater than the robust LM value of error. SEM also had the lowest AIC value, and highest log likelihood value (explaining better model suitability), which led us to use it to investigate the spatial dependence of exclusive use of hygienic materials among urban women.

Results of the SEM are also presented in Table 4. The lambda value was 0.54 ($p < 0.001$), which is highly significant and implies that the outcome variable is spatially dependent on neighbouring values. SEM confirmed that respondent's years of schooling ($\beta$: 0.055, $p < 0.001$), marital status ($\beta$: 0.262, $p$: 0.012), social groups ($\beta$: 0.158, $p < 0.001$), and household wealth status ($\beta$: −0.265, $p < 0.001$) were statistically significant indicators of exclusive use of hygienic materials among urban women. In other words, SEM model depicts that if in a district, among urban women, child marriage (marriage before 18 years) increased by 10%, then the exclusive use of hygienic materials significantly decreased by 2.6%. Similarly, if in a particular district, the proportion of poor women increased by 10%, then the exclusive use of hygienic materials decreased by 2.6%.
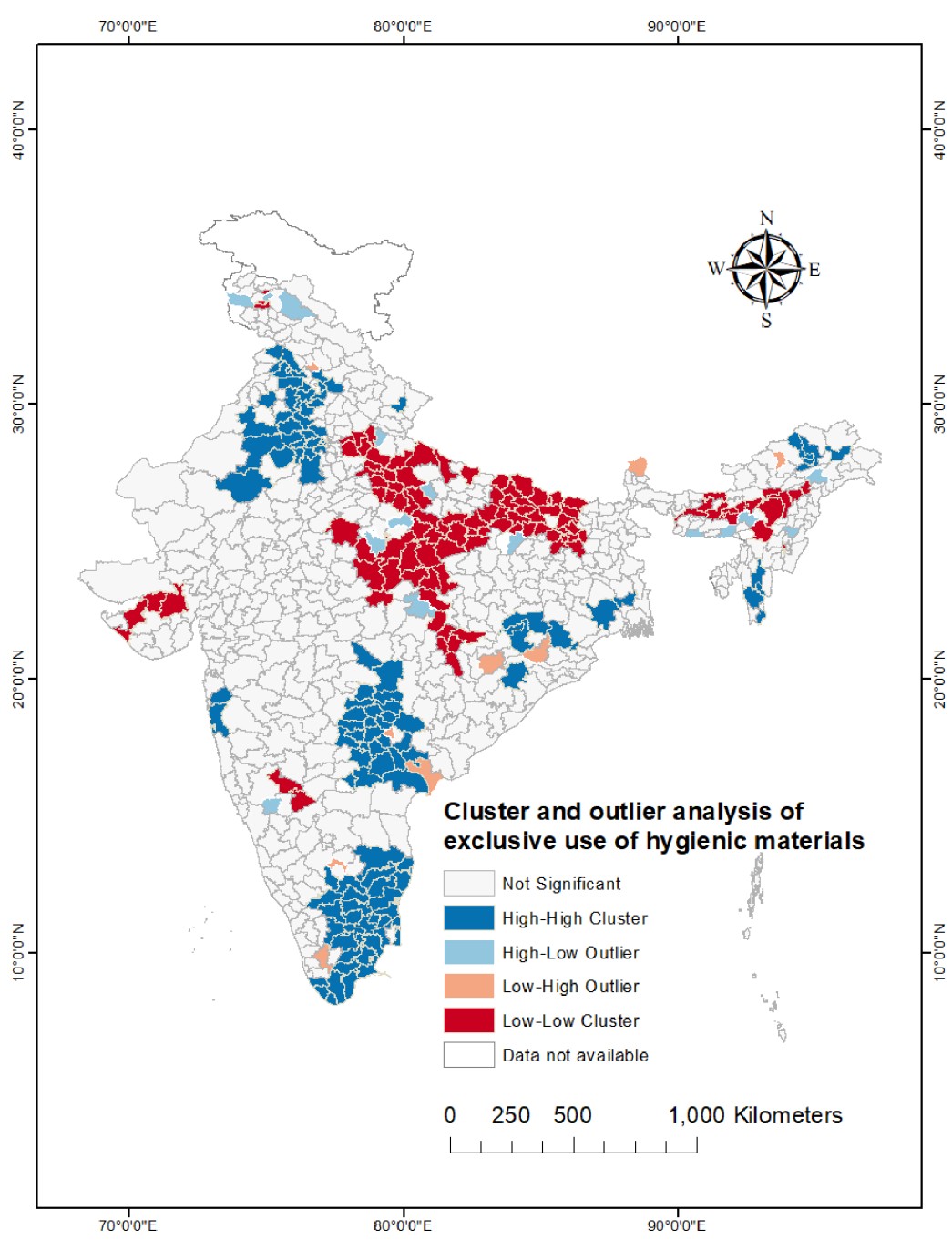

**Figure 4** Cluster and outlier analysis map (Anselin Local Moran's I) showing the statistically significant (*p* value < 0.05) spatial clusters and outliers of exclusive use of hygienic materials among urban women (15–24 years) across the districts of India.

## DISCUSSION

According to the 'urban advantage theory', urban women have a health and healthcare use advantage over rural women. Compared to rural India, the exclusive use of hygienic

**Table 4 Spatial regression model for estimating spatial association between percentage of exclusive use of hygienic absorbents among urban women (15–24 years) and background characteristics, NFHS-5.**

| Variables | OLS | SLM | SEM |
|---|---|---|---|
| Age of respondent (in years) | −0.019 (<0.001) | −0.011 (0.031) | −0.003 (0.643) |
| Age at menarche (in years) | −0.000 (0.923) | 0.001 (0.713) | 0.005 (0.148) |
| Years of schooling | 0.077 (<0.001) | 0.065 (<0.001) | 0.055 (<0.001) |
| Marital status | −0.630(<0.001) | −0.443 (<0.001) | −0.262 (0.012) |
| Hindu | 0.038 (0.110) | 0.020 (0.378) | 0.027 (0.308) |
| SC/ST | 0.245 (<0.001) | 0.192 (<0.001) | 0.158 (<0.001) |
| Poor | −0.268 (<0.001) | −0.214 (<0.001) | −0.265 (<0.001) |
| Mass media exposure | 0.207 (0.009) | 0.135 (0.065) | 0.086 (0.254) |
| Rho | | 0.342 (<0.001) | |
| Lambda | | | 0.542 (<0.001) |
| AIC | −631.17 | −722.93 | −754.08 |
| SW | −590.09 | −677.30 | −713.00 |
| Log Likelihood | 324.59 | 371.47 | 386.04 |

**Notes.**
OLS, Ordinary least squares; SLM, Spatial lag model; SEM, Spatial error model; AIC, Akaike information criterion; SW, Schwarz criterion.

materials by young women in urban India is far greater. However, the above statement is only partly accurate, as our research indicated large socioeconomic, biodemographic, and geographical disparities in the exclusive use of hygienic materials among urban women. The purpose of this research was to discover spatial heterogeneities in the exclusive use of hygienic materials among young urban women in India by district. Our research revealed that the exclusive use of hygienic materials was not spread equally throughout the districts of urban India. As indicated by the district-wise prevalence map, there is a large disparity between the exclusive use of hygienic materials in the urban areas of central India and southern India. This result further corroborated by our spatial analysis at district level. Global Moran's I, a measure of geographical autocorrelation, found a considerable clustering in the exclusive use of hygienic materials among urban India's young women. In addition, cluster and outlier analyses were used to pinpoint the precise location of clusters of exclusive use of hygienic materials. In addition, geographical regressions were performed to investigate the relevant parameters influencing the exclusive use among young urban Indian women.

Any efforts to extend the reach of safe, affordable, and hygienic menstrual materials exploration of problematic geographical clusters with very poor prevalence of exclusive use of hygienic materials will be first essential step, further identification of the vulnerable groups of women within urban areas can help in developing context-specific programs and strategies. In this context, our research identified a number of decisive findings.

Firstly, high level clustering was discovered in the districts of Uttar Pradesh, Bihar, and Madhya Pradesh, Chhattisgarh, and in most of the north-eastern states (except Arunachal Pradesh and Mizoram) where the exclusive use of hygienic materials was very low. It is worth mentioning here that, menstruation carries a greater social stigma in districts of Madhya Pradesh and Uttar Pradesh, Bihar and Chhattisgarh, and particularly in India's

north-eastern states (*McCammon et al., 2020*; *Sopam, 2021*). These states also have low socioeconomic development, educational attainment, and healthcare access (*Arokiasamy & Gautam, 2008*; *Arokiasamy et al., 2013*). As a result, urban women are less likely to utilize sanitary materials in these states (*Ram et al., 2020*; *Roy et al., 2021*). It is essential to focus in the pockets of ultra-low (less than 30%) exclusive use of hygienic materials mainly in the districts of Ambedkar Nagar, Hardoi, Ballia, Shrawasti, Mau, Bara Bnaki, Sant Kabir Nagar, Sitapur, Hamirpur of Uttar Pradesh; Anantnag, Kupwara, Badgam, Baramula of Jammu & Kashmir; Kaimur (Bhabua), Sitamarhi, Gopalganj, Nawada of Bihar; Bemetara, and Raipur of Chhattisgarh, if overall prevalence of exclusive use of hygienic materials among urban women is to be increased.

The higher prevalence of exclusive use of hygienic materials among young urban women in the southern states of Andhra Pradesh, Tamil Nadu, and Telangana can be attributed to the successful implementation of free and subsidized sanitary napkin programs. A case in point is the "Pudhu Yugam" scheme in Tamil Nadu, which provides women and girls with a monthly supply of free sanitary napkins (*Tamil Nadu Urban Sanitation Support Programme, 2022*). These states have also taken proactive steps to promote the use of hygienic materials, such as installing soap dispensers and napkin vending machines in schools in collaboration with local non-governmental organizations (*Geertz et al., 2016*; *National Rural Health Mission, 2016*). The hygienic materials usage in the north-eastern region of Karnataka appears to lag behind other parts of the state. This disparity could be attributed to the region's lack of development in comparison (*Rajanna, 2022*). Yet, to uncover the root cause, a deeper delve into the matter is imperative

Secondly, women's years of schooling, social groups, household wealth status, and marital status emerged as critical determinants of the exclusive use of hygienic materials. Previous studies have documented that the prevalence of exclusive hygienic materials is relatively low among SC/ST women (*Anand, Singh & Unisa, 2015*; *Ram et al., 2020*; *Roy et al., 2021*). Our results, confirming the same, clearly indicate that urban SC/ST women are less likely to exclusively use hygienic materials during their menstruation. In addition to a lack of awareness and affordability, discrimination in healthcare facilities may be behind these groups' low use of hygienic materials (*Roy et al., 2021*).

With the increase in household wealth status, the exclusive use of hygienic materials has increased significantly (*Ram et al., 2020*; *Chauhan et al., 2021*; *Roy et al., 2021*). Poverty, along with social taboo and awareness (of the schemes provisioning free sanitary pads) has been identified as a major risk factor and one of the major barriers which may create an obstacle to the exclusive use of hygienic materials (*Roy et al., 2021*). In India, the average price of a sanitary napkins ranges from 5 to 12 rupees, making them a luxury item for the improverished women in urban India (*Raghavan, 2018*; *Rodriguez, 2019*).

The likelihood of exclusive use of hygienic materials increased with the increasing years of schooling of the urban women. Previous studies have noted that higher educated women are usually well-versed in using hygienic materials (*Roy et al., 2021*) and are aware of the dangers of unsanitary menstrual traditions (*Sonowal & Talukdar, 2019*).

The present study revealed that the exclusive use of hygienic materials was inversely associated with child marriage among urban women. Age at marriage positively relates
with educational attainment and awareness of hygienic materials (*Surendran et al., 2021*; *Maharana, 2022*). Similarly, women's autonomy in making decisions and managing their finances improves with age (*Dutra-Thomé et al., 2019*). Thus, these are some potential causes of increase in the use of hygienic material with increasing age at marriage.

The present study has some limitations. Firstly, we could not include some variables like ownership of mobile phone and bank account, respondent's autonomy in our study as in the NFHS-5, these questions were asked to currently married women only. Secondly, the NFHS-5 is a cross-sectional dataset, and therefore, it was not possible to establish causal relationship between predictors and outcome variables. Thirdly, as the NFHS-5 does not provide any supply side variables which, according to the demand-supply framework of healthcare utilization, are important in explaining the level of utilisation of a service or product. Another limitation of the study lies in the fact that sample sizes for some districts are not adequate. This means that the estimates for these districts have wider confidence intervals, requiring the reader to exercise caution when interpreting the results. Lastly, in the NFHS-5 the menstrual hygiene questions were asked only to women aged 15–24, so we could not include other menstruating women in our study.

## CONCLUSION

The study demonstrates that exclusive use of hygienic materials among urban women varies considerably across the districts of India. A cluster of districts with low use of hygienic materials is located in central states of India demanding immediate policy attention. In many of these districts, low use of hygienic materials among urban women coexists with high poverty rates, low female education, low marriage age, and a high proportion of SC/ST population. A contextual replication of policy and programs that have been successful in raising the level of hygienic materials use in some southern districts or states may help in raising the usage in states/districts with low exclusive use of hygienic materials.

## APPENDIX

**Spatial autocorrelation**

Spatial autocorrelation generally measures the degree of correlation between the value of a variable in a specific location and the values of the same variable at neighbouring locations.

**Global Moran's I**

Global Moran's I estimates spatial autocorrelation by the single value for the entire study area, known as global spatial autocorrelation measures (*Tsai et al., 2009*).

Global Moran's I index: $I = \dfrac{n}{\sum_i^n \sum_i^n w_{ij}} \dfrac{\sum_i^n \sum_i^n w_{ij}(x_i - \overline{x})(x_j - \overline{x})}{\sum_i^n (x_i - \overline{x})^2}$.

Where $I$ is Moran's I value, n is the number of the spatial features, $x_i$ is the attribute value of feature I, $x_j$ is the attribute value of feature j, $\overline{x}$ is the mean of this attribute, $w_{i,j}$ is the spatial weight between feature i and j, $\sum_i^n \sum_i^n w_{ij}$ is the aggregation of all spatial weights. The tool calculated the mean $\overline{x}$, the deviation from the mean $(x_i - \overline{x})$ and the data variance

$\frac{\sum_{i=1}^{n}(x_i-\overline{x})^2}{n}$ (denominator). Deviations from all neighbouring features are multiplied to create cross-products (the covariance term). Then, the covariance term is multiplied by the spatial weight. All other parameters are used to normalize the value of the index. Moran's-I values vary from −1 to +1. A value of zero denotes a random spatial pattern. Positive Moran's I value indicates the clustering in the geographical distribution and positive spatial autocorrelation (*i.e.,* nearby locations have similar values) and vice versa.

**Local Moran's I**

Local Moran's I index estimates spatial autocorrelation at the local level and identifies clustering of high or low values and also traces spatial outliers (*Anselin, 1995*).The formula to calculate the Local Moran's I statistic is as follows:

$$\text{Local Moran's I}: I_i = \frac{x_i - \overline{X}}{m_2} \sum_j w_{ij}(x_j - \overline{X}); m_2 = \frac{\sum_i (x_j - \overline{X})}{n-1}.$$

Where n is the total number of observations (spatial objects), $x_i$ is the attribute value feature j, $x_j$ is the attribute value of feature j, $\overline{X}$ is the mean of this attribute, $w_{i,j}$ is the spatial weight between feature i and j, $m_2$ is a constant for all locations. It is a consistent but not unbiased estimate of the variance.

### Funding
This work was supported by the Institute of Eminence (IoE) Seed Grant provided by Banaras Hindu University (No. R/Dev/D/IoE/Equipment/Seed Grant-II/2022-23/48726). The funders had no role in study design, data collection and analysis, decision to publish, or preparation of the manuscript.

### Grant Disclosures
The following grant information was disclosed by the authors:
Institute of Eminence (IoE) Seed Grant: R/Dev/D/IoE/Equipment/Seed Grant-II/2022-23/48726.

### Competing Interests
The authors declare there are no competing interests.

### Author Contributions
- Aditya Singh conceived and designed the experiments, performed the experiments, analyzed the data, prepared figures and/or tables, authored or reviewed drafts of the article, and approved the final draft.
- Mahashweta Chakrabarty conceived and designed the experiments, performed the experiments, analyzed the data, prepared figures and/or tables, authored or reviewed drafts of the article, and approved the final draft.

### Data Availability
The National Family Health Survey-5 dataset used in the study is available at the official website of Demographic and Health Surveys (DHS): https://dhsprogram.com/data/available-datasets.cfm.

This data can be obtained by registering as a DHS data user and requesting access for legitimate research purposes: https://dhsprogram.com/data/Access-Instructions.cfm.

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
