# Peer review of "Spatial heterogeneity in the exclusive use of hygienic materials during menstruation among women in urban India"

_PeerJ, doi:10.7717/peerj.15026_

## Round 0.1 · original submission · Major Revisions

This manuscript details the investigation aimed to document disparities in menstrual hygiene and identify vulnerable groups for the same. This secondary data analysis was done on nationally representative dataset of NFHS-5.
Three reviewers have given their comments and authors are advised to modify the manuscript as per suggestions.
Additionally issues are as follows
1) Methods section doesn't have the flow of observational epidemiological studies. Organise methods section accordingly. Provide details in each sub-section of methods as well.
2) Considering the nature of sampling of NFHS-5 wherein clusters are selected in multiple stages. Within each district few clusters are selected. Considering this whether cluster analysis using Local index can be done, needs to be explained.
3) Only urban clusters are analysed, what is the background probability of clustering of urban areas? Whether this fact needs to be adjusted in the spatial analysis?
3) How robust is the estimate at district level? You have chosen to restrict on urban sample that too in 15-24 age group and thus few district may have it in only two digits and very few will have it more than 300. How valid it to label the districts under different categories of prevalence. This is the critical issue when we go at granular level with filters of urban and 15-24 age group and then extrapolate at district level.
4) Whether regression analyses were wighted? If not, reanalyse using weights. If yes mention it.
5) Limitations as pointed above and by reviewers to be included.

·

Basic reporting

No comment

Experimental design

In table 1 it is written that “The age of marriage variable in the NFHS-5 data has been classified into two categories. Women who married before the age of 18 are referred to as child marriage (1), whereas those who married after the age of 18 are referred to as non-child marriage (0).” Please explain what was done to the women who were below 18 years (N = 26,509) and not married (approx. 90% of N)

Accordingly all the results and discussion related to child marriage needs to be corrected.

Mention as limitation of data source that data represents only urban women aged 15-24

Please provide reference to the details of Global and Local Moran’s I index and Lagrange Multiplier (LM) and Robust Lagrange Multiplier (LM) diagnostic tests

Please provide details of user licenses for proprietary software used e.g. ArcGIS, Stata16 and others if any

Validity of the findings

In the first paragraph of results term "significant" has been used for "proportion of urban women did not discuss menstrual hygiene with healthcare workers three months before the study"
significant is a statistical term please explain statistical evidence how it is significant? or replace the term

Paragraph 4 of

"349 out of 707 districts or almost 50% of Indian districts mainly from the sates of Uttar Pradesh, Bihar, Chhattisgarh, western Gujarat and some North-eastern states like Meghalaya, Assam, Nagaland, Manipur have lower exclusive use of hygienic absorbents than the national average (66.8%). >

results needs to reframed because in a normal distribution it is well expected that 50% districts will have values above national average.

Additional comments

Some grammatical errors like repeated words needs to be corrected

Reviewer 2 ·

Basic reporting

Reviewer Comments 
I want to thank the editors for providing me with an opportunity to review this paper. I would also like to congratulate the authors for choosing a crucial topic for the research, i.e., Spatial heterogeneity in the exclusive use of hygienic absorbents during menstruation among urban women in India.
Overall, it is a decent work and advances the scholarship. I have added some of the comments below to improve the paper. 

Introduction
I really like the introduction, however, you can strengthen it by showing how managing menstruation is a human rights issue (Hennegan, 2017; Babbar et al., 2022) and sustainable development goals (Sommer et al., 2022). 

I would also encourage you to add more recent literature on your argument "Inadequate access to toilets, sanitation, water, and drainage, as well as growing poverty and a lack of disposable income in cities, makes living conditions in Indian cities unsanitary and unhygienic." You've mentioned a 2016 paper and it might be helpful to discuss how the world has evolved out of COVID and how it had larger implications on access to WASH facilities. 

You're missing multiple important studies while discussing this argument "t is reported that there are significant differences in women's choice of menstrual absorbents based on region, socio-economic status, and demographic variables in India."  All of these studies have discussed the women's choice of menstrual absorbents either at India level or at urban and rural India level. You may want to change the statement mentioned above and incorporate the given studies and claim that none of these studies have looked in spatial heterogeneity of the menstrual products. 
1. Anand, E., Singh, J. and Unisa, S. (2015) ‘Menstrual hygiene practices and its association with reproductive tract infections and abnormal vaginal discharge among women in India’, Sexual and Reproductive Healthcare 6(4): 249–254. https://doi.org/10.1016/j.srhc.2015.06.001 
2. Almeida-Velasco, A. and Sivakami, M. (2019) ‘Menstrual hygiene management and reproductive tract infections: a comparison between rural and urban India’, Waterlines 38(2): 94–112. https://doi.org/10.3362/1756-3488.18-00032.
3. Babbar, K., Saluja, D., & Sivakami, M. (2021) 'How socio-demographic and mass media factors affect sanitary item usage among women in rural and urban India', Waterlines 40(3): 160-178. https://doi.org/10.3362/1756-3488.21-00003
4. Goli, S., Sharif, N., Paul, S. and Salve, P.S. (2020) ‘Geographical disparity and socio-demographic correlates of menstrual absorbent use in India: a cross-sectional study of girls aged 15–24 years’, Children and Youth Services Review 117: 105283 https://doi.org/10.1016/j.childyouth.2020.105283.
5. Roy, A., Paul, P., Saha, J., Barman, B., Kapasia, N. and Chouhan, P. (2020) ‘Prevalence and correlates of menstrual hygiene practices among young currently married women aged 15–24 years: an analysis from a nationally representative survey of India’, European Journal of Contraception and Reproductive Health Care 26(1): 1–10. https://doi.org/10.1080/13625187.2020.1810227.

While finishing the introduction, I just could not figure out why we need to focus especially on urban women? You may want to summarise it in 1-2 lines before you end your introduction.

Experimental design

Methods
May I know why you did not consider the menstrual cups as a hygienic absorbent? Were there any specific reasons for not including it in the list?

In the data analysis section, you mention "Both the LM and Robust LM tests for Spatial Lag Model (SLM) and Spatial Error Model (SEM) were found to be significant. Since the SLM had the lowest Akaike Information Criterion (AIC) value, highest log-likelihood value, highest LM, and Robust LM values, we estimated it to be the best model out of the three. ArcGIS 10.5, GeoDa, GeoDaSpace, and Stata16 were utilized to analyse the data for this study." This reads more like what you've seen in the results and not generally how it should flow? This information should be presented in the results section with the explanation on why OLS, SLM and SEM was conducted. Data analysis section should explain the methodology in general.

You've mentioned a bunch of control variables "current age of respondents, age at menarche, child marriage, respondentís years of schooling, religion, social groups, household wealth status, type of home, mass media exposure, discussion on menstrual hygiene with healthcare workers." You need to explain in detail why these are used as control variables and a mere one line of explanation "The choice of variables is guided by existing literature on menstrual hygiene management (18ñ21,24,25)." may not help.

Why is age at menarche used as a binary variable i.e., Less than 13 and more than 13? Is there a rationale behind this?

How is the wealth index constructed as poor vs non-poor? Any rationale behind this? You're missing out on a lot of information as DHS defines it as poorest, poor, middle, rich and richest.

Similarly, you can show more variations in terms of religion and caste. I am sure there is enough variation to show Hindu, Muslim and others. Similarly, for caste, you can show General, OBC and SC/ST. Is there any rationale why you've sub-divided them into two categories only? Also, is it supported by previous literature?

I would strongly encourage you to put a paragraph on how these variables are constructed. I understand that you've added this in Table 1, however, a paragraph here may help the reader in understanding the measures better.

For your last variable, discussion of MH with CHW, the variation in the responses is quite small. Less than 2% of the respondents said yes. It might be good to remove this variable 

Results
Overall, I get the impression that a lot of importance is given to the technical part of the statistics. I fully understand and appreciate your grasp on that. However,  what value these statistics add to answer your research questions seem missing.  This seems to be missing at least from my perspective. For instance, why do we need to do spatial auto-correlation?

Similarly, results can be written in a more coherent manner. For instance, one single line "Coefficient of child marriage was found to be highest (β=-0.37, p value< 0.00), followed by poor household status (β=-0.19, p value< 0.00), SC/ST (β=0.19, p value< 0.00), and years of schooling (β=0.05, p value< 0.00)" won't be sufficient to describe your overall results and needs much more deliberation. 

Additionally, I have suggested multiple changes in the measures section of the methodology. I would request you to incorporate those changes and update the results accordingly.

Validity of the findings

Discussion
In my mind, the discussion section starts with brief outlining of key research questions, brief results, contribution of the study. These parts are currently missing from the first paragraph of the discussion section. It should clearly outline the story of what we have done in the research paper till now, which seems to be missing. 

Your first finding talks about "Firstly, our study identified a significant geographical inequity in the district-level prevalence of using only hygienic absorbents, where the prevalence gap was as high as 90%. Only 10% of urban women in Kaimur district of Bihar exclusively use hygienic absorbents, whereas this prevalence rate was 100% in the Siddipet district of Telangana." But you do not mention the reasons behind this inequity.

In the second paragraph of the discussion section, you mention "It is worth mentioning here that, menstruation carries a greater social stigma in districts of Madhya Pradesh and Uttar Pradesh, Bihar and Chhattisgarh, and particularly in India's north-eastern states. These states also have low socioeconomic development, educational attainment, and healthcare access." These arguments were made with no references. Pls cite some recent studies to show this. 

In the next paragraph, again you talk about "In the same geographical zone, evidently lower224 usage of hygienic absorbents in a cluster of districts in north-eastern Karnataka may be due to its225 relative underdevelopment as compared with rest of Karnataka" without any references. 

How are the importance of the control variables in this particular statement identified? "Thirdly, in order of importance (from most significant to least), social groups, household wealth status, years of schooling, and child marriage emerged as critical determinants of the exclusive use of hygienic absorbents." Is it on the basis of the coefficients? Is this the right away to identify the most important variables? In my mind, one should use decomposition analysis to understand which variables contribute the highest percentage to the usage of menstrual absorbents?

You cite a 2011 study to show "A pack of 10 sanitary napkins costs around 30ñ40 INR (0.39-0.52 USD), which is240 unaffordable for poor households in India (29)." I am sure it costs a lot more now. Pls use a recent study to further explain this. 

When you discuss the usage of menstrual absorbents and child marriage, you make multiple statements without any references. Pls look into it and add references.

You may want to add "not" in your second limitation.

·

Basic reporting

Nice attempt on the part of the authors to bring about a pertinent topic. However, there are a few concerns. Language is not adequately proofread. Language, grammar, and punctuation can be improved in a number of places in the manuscript; for example, in line 263 there is repeat usage of a word, ".....of of.....", in line 254 the authors probably mean "...not possible....." and the "not" seems to be missing. I would suggest contacting a professional editing service.

Experimental design

The Introduction is intelligently drafted with a nice build-up. The novelty, knowledge gap, and research question are relevant and well-defined.
One of the concerns is the mention of an Ethical statement. I understand that the data is taken from an open domain. The authors can include a statement supporting the same in the manuscript like ..." Ethical permission was not sought as ..............".

Validity of the findings

STROBE checklist can be included by the authors.

Additional comments

Abbreviations have been used at a few places in the manuscript at the first usage, please avoid the same.
For example use of "SLM" in the Abstract section.
Use of "OLS" in line 94, and use of "SC/ST" in line 122.

---

## Round 0.2 · accepted · Accept

Suggestions of the reviewers and the editor have now been incorporated. Manuscript is thoroughly revised and ready for publication.
Congratulations!!!

Reviewer 2 ·

Basic reporting

I would like to congratulate the authors for the revision version of the manuscript. This article looks good to be published now.

Experimental design

It looks good.

Validity of the findings

It looks good.